# Inspiratory Muscle Performance and Its Correlates Among Division I American Football Players

**DOI:** 10.3390/jfmk10040470

**Published:** 2025-12-04

**Authors:** Luis A. Feigenbaum, Lawrence P. Cahalin, Jeffrey T. Ruiz, Tristen Asken, Meryl I. Cohen, Vincent A. Scavo, Lee D. Kaplan, Julia L. Rapicavoli

**Affiliations:** 1Department of Physical Therapy, Miller School of Medicine, University of Miami, Coral Gables, FL 33146, USA; l.cahalin@miami.edu (L.P.C.); mcohen@miami.edu (M.I.C.); jrapicavoli@med.miami.edu (J.L.R.); 2Department of Athletics, Physical Therapy, University of Miami, Coral Gables, FL 33146, USA; 3UHealth Sports Medicine Institute, University of Miami, Miami, FL 33136, USA

**Keywords:** inspiratory muscle strength, Division I football physiology, anthropometrics, respiratory endurance, TIRE protocol, positional differences

## Abstract

**Background**: Inspiratory muscle performance plays a crucial role in athletic demands, yet its associations with anthropometric and positional variables in American football remain underexplored. This study examined relationships between inspiratory metrics and key characteristics in Division I collegiate football players. **Methods**: Eighty-five Division I collegiate football players (mean academic year in school: 2.87; height: 74.3 inches; weight: 108.13 kg; BMI: 30.21) underwent the Test of Incremental Respiratory Endurance (TIRE) to measure maximal inspiratory pressure (MIP), sustained maximal inspiratory pressure (SMIP), and inspiratory duration (ID). Bivariate and multivariate analyses assessed associations with height, weight, BMI, year in school, offense/defense status, and playing position. **Results**: Inspiratory performance showed limited associations with anthropometric variables, with only a weak correlation between height and ID. (ρ = 0.243, *p* = 0.024). No significant differences were observed by year in school or position **Conclusions**: Inspiratory performance appears largely independent of anthropometric and positional factors; future research should explore targeted respiratory training as a hypothesis rather than a confirmed benefit.

## 1. Introduction

American football is a physically demanding sport characterized by short bursts of high-intensity activity, frequent collisions, and position-specific physiological requirements [1]. At the collegiate level, athletes are often stratified by position into categories such as linemen and skill players, each with distinct anthropometric and performance profiles [2,3,4]. While researchers have examined strength, speed, and injury risk in football players, relatively little attention has been paid to the role of respiratory muscle performance, particularly inspiratory strength, in this population [5,6,7].

Inspiratory muscle performance, typically measured via maximal inspiratory pressure (MIP), sustained maximal inspiratory pressure (SMIP), and inspiratory duration (ID), reflects the strength and endurance of the diaphragm and accessory respiratory muscles. These parameters are increasingly recognized as relevant to athletic performance, especially in sports requiring repeated high-intensity efforts and rapid recovery [8,9,10]. Fogarty et al. provided a detailed overview of diaphragm motor unit recruitment and control, emphasizing the size principle where fatigue-resistant motor units are recruited first during low-force ventilatory behaviors [11]. This framework is relevant to football, where players engage in short bursts of maximal effort. The ability to recruit higher-force diaphragm motor units efficiently may contribute to better oxygen delivery and reduced fatigue, especially in linemen during repeated sprints or tackles, as well as core strength.

Emerging evidence suggests that inspiratory muscle strength may correlate with anthropometric and demographic variables such as age, body mass index (BMI), height, and playing position. Karaduman et al. found that athletes with BMIs in the 18.5–29.9 kg/m^2^ range exhibited significantly higher MIP and pulmonary function scores compared to those in underweight or obese categories [12]. This suggests an optimal BMI range for respiratory efficiency, potentially relevant for football players whose body composition varies widely by position.

The diaphragm plays a pivotal role in posture by generating intra-abdominal pressure (IAP) during descent, which stiffens the lumbar spine and pelvis through co-contraction with the transversus abdominis, pelvic floor, and multifidus, forming the core stability canister [11]. This mechanism supports anticipatory postural adjustments prior to limb movements and modulates balance in standing tasks [11]. Respiratory muscle training, particularly inspiratory muscle training, strengthens diaphragmatic function, enhancing IAP generation and core muscle synergy, which may indirectly improve trunk stability and balance beyond respiratory benefits, with implications for athletic performance and injury prevention [10,12].

Despite growing interest in respiratory muscle training and performance, few studies have systematically examined the correlates of inspiratory performance in American football players. Most existing research focuses on endurance athletes or general athletic populations, leaving a gap in understanding how inspiratory strength interacts with age, anthropometry, and positional roles in football.

The purpose of this study is to evaluate the relationship between inspiratory muscle performance and key demographic and anthropometric variables in Division I collegiate football players (D1CFP). By identifying potential correlates of inspiratory performance, this research aims to inform training strategies and contribute to a more comprehensive understanding of performance determinants in collegiate football players. The research team predicts that inspiratory performance will be positively correlated with height and weight, will vary by playing position, and will improve with years in college. Finally, the team hypothesizes that the relationship between inspiratory performance and BMI will demonstrate a negative correlation.

We hypothesized that inspiratory performance would correlate with height and weight, vary by position, and improve with years in college. We also anticipated a negative correlation with BMI, acknowledging BMI as an imperfect proxy for body composition in football players

## 2. Materials and Methods

### 2.1. Participants

Eighty-five male D1CFP from the same team participated in this study. Positional classification was based on the official team depth chart at the time of testing. Due to small sample sizes in certain positions (e.g., quarterbacks), findings for these groups should be interpreted with caution. A power analysis was conducted to determine the sample size needed to detect significant correlations between inspiratory performance (MIP, SMIP, ID), anthropometric variables (height, weight, BMI), and sport-related variables (position and year in school) in a group of D1CFP. Based on prior studies suggesting moderate-to-large correlations in athletic populations [12], we assumed a moderate effect size (r = 0.3) for Spearman correlations, with a significance level of α = 0.05 and power of 0.80. This yielded a minimum sample size of approximately 84. The current sample of 85 participants meets this threshold, providing adequate power to detect moderate effects in bivariate analyses. For multivariate regression analyses, power was not separately calculated due to the exploratory nature of the study, but the sample size supports detecting moderate-to-large effects (R^2^ ≈ 0.13) with 15 predictors.

Inclusion criteria included being a D1CFP and being free from respiratory or musculoskeletal conditions that could impair normal function. Exclusion criteria included recent injuries (within the past 6 months) or respiratory illnesses. Participants were recruited through convenience sampling from a single D1CF team.

### 2.2. Procedures

Participants provided informed consent prior to study participation, and the research adhered to the principles of the Declaration of Helsinki. The study protocol received approval from the University’s institutional ethics committee, and no external funding was obtained for this research. Testing occurred during the pre-season in a controlled environment (22–24 °C, 50–60% humidity). Participants completed a standardized 10 min warm-up consisting of dynamic hamstring and quadriceps stretches, trunk rotations, and shoulder mobility drills.

### 2.3. Test of Incremental Respiratory Endurance (TIRE)

The TIRE protocol used the RT2 device (DeVilbiss Healthcare Ltd., Wollaston, UK), calibrated before each session per manufacturer guidelines by a single trained assessor (a physical therapist specializing in cardiopulmonary physical therapy with over 30 years of experience). Participants were familiarized with the protocol via a practice trial. Instructions on the testing were to exhale fully, then inhaled forcefully to obtain the MIP at 1–2 s of inspiration and to sustain the effort for as long as possible, prompted by “inspire deep, hard, and long” cues. Three to five trials were conducted in a seated position, with the highest MIP and SMIP trial selected (prioritizing MIP). Verbal encouragement and real-time biofeedback were provided via TIRE software (v2.52). SMIP was measured in Pressure-Time Units (PTU), representing the area under the inspiratory pressure-time curve (Figure 1). Participants rested 60–90 s between trials; testing stopped after three successful trials or participant fatigue.

As shown in Figure 1, MIP indicates the highest pressure generated during the first second of an inspiratory breath. SMIP quantifies the cumulative pressure produced over the entire duration of a sustained inhalation, providing insights into the strength, endurance, and work capacity of these muscles. Inspiratory duration (ID) measures the length of sustained inhalation in seconds.

### 2.4. Statistical Analyses

All statistical analyses were conducted using SPSS v28, with the significance level set at *p* < 0.05. Descriptive statistics were calculated for all variables, including means and standard deviations. Given the exploratory nature and multiple comparisons (~30 tests), *p*-values were unadjusted. Readers should interpret borderline results (e.g., *p* = 0.024) cautiously, as they may not remain significant after correction. MIP was normally distributed and compared using *t*-tests, while SMIP and ID were analyzed with non-parametric tests. The Shapiro–Wilk test was used to assess normality for continuous variables. Since most variables (except height and MIP) violated normality assumptions based on Shapiro–Wilk tests (*p* < 0.05), non-parametric tests were prioritized for bivariate analyses: Spearman correlations for continuous/ordinal variables, Mann–Whitney U for binary groups (offense/defense), and Kruskal–Wallis for multi-group comparisons (positions). Given the exploratory nature of the study, *p*-values were unadjusted, but readers should interpret borderline results cautiously due to multiple comparisons (with approximately 30 tests, a Bonferroni-corrected alpha of ≈0.0017 would negate borderline results). Effect sizes for correlations were interpreted as weak (|ρ| < 0.3), moderate (|ρ| = 0.3–0.5), or strong (|ρ| > 0.5). Cohen’s d was calculated for between-group comparisons (small: 0.2, medium: 0.5, large: 0.8).

For multivariate analyses, three separate multiple linear regression models were constructed for MIP, SMIP, and ID as dependent variables. To avoid multicollinearity (variance inflation factors > 10 for BMI due to its derivation from height and weight), BMI was excluded as a predictor. Each model included 11 predictors: height, weight, year in school (continuous), offense/defense status (binary: 0 = offense, 1 = defense), and seven dummy coded position categories (with quarterback as reference). Residuals were assessed for normality using Q-Q plots and Shapiro–Wilk tests; no significant violations were found (*p* > 0.05). Due to small subgroup sizes (e.g., kicker: n = 1), the kicker was excluded from position-based analyses to improve estimate stability. Variance inflation factors (VIF) were computed; BMI was excluded due to VIF >10. Position was dummy-coded into 7 levels.

## 3. Results

### 3.1. Descriptive Statistics

The final dataset comprised 85 D1CFPs with complete data across all variables. Descriptive statistics are presented in Table 1.

Inspiratory muscle performance by positional group included 46 offensive and 39 defensive players (Table 2).

### 3.2. Bivariate Analyses

Normality testing revealed that only height and MIP were normally distributed; all other variables violated normality assumptions. Consequently, non-parametric tests were prioritized.

#### 3.2.1. Results of Correlation Analyses of Inspiratory Muscle Performance with Anthropometrics

Spearman correlation analyses showed no significant associations between weight or BMI and any inspiratory metric. Height and ID demonstrated a weak positive correlation (ρ = 0.243, *p* = 0.024) (Table 3).

#### 3.2.2. Results of Correlation Analyses Between Inspiratory Muscle Performance and Year in School

No significant correlations were observed between inspiratory muscle performance and year in school, but a trend towards a relationship between MIP and year in school was found (Table 4).

#### 3.2.3. Comparison of Inspiratory Performance Between Offensive vs. Defensive Players

No significant differences between offensive and defensive players were observed (Table 5).

## 4. Discussion

This study found limited associations between inspiratory performance and anthropometric variables, with only a weak correlation between height and inspiratory duration. Positional differences were modest, with linemen and tight ends showing higher SMIP and ID compared to skill positions. Bivariate analyses revealed limited associations between inspiratory performance and height, but not weight, BMI, or year in school. Additionally, no statistically significant differences in inspiratory performance were found between position groups. Multivariate regression models identified modest predictors of inspiratory metrics, particularly among linemen and more experienced players. These findings suggest that inspiratory performance may not be inherently tied to anthropometric traits or positional classification alone, but rather influenced by training history, neuromuscular control, and acute physiological interventions. Regression models yielded low R^2^ values, indicating that unmeasured factors such as training history, respiratory conditioning, and genetic predispositions likely play a major role in inspiratory performance.

Longitudinal data from NCAA Division I football players also suggest that physical development over a collegiate career influences performance metrics. Jacobson et al. observed significant improvements in strength and power measures across years in school, with skill players showing early gains in vertical jump and linemen maintaining consistent size and strength [13,14]. These findings support the inclusion of “year in college” as a variable potentially associated with inspiratory performance.

### 4.1. Inspiratory Muscle Performance and Football-Specific Demands

Our findings are partially aligned with those of Abdulrahman and Mahmood, who reported no significant correlation between body composition and respiratory muscle strength in a sample of football players [15]. Although our results showed a weak positive correlation between height and inspiratory duration, further research is needed to clarify whether anthropometric factors influence inspiratory performance.

With respect to positional differences in inspiratory performance, our results align with and extend prior research emphasizing position-specific physiological adaptations in American football. Robertson et al. profiled NCAA Division II football starters and reported significant between-position differences in body composition, lower-body power, and maximal strength, with linemen demonstrating greater fat-free mass and absolute strength, while skill players exhibited superior power-to-weight ratios [16]. These findings demonstrate the divergent training and biomechanical demands across positions: linemen engage in repeated maximal-effort isometric and concentric contractions during blocking and tackling, whereas skill positions rely on explosive speed and agility. Our study complements this framework by revealing that linemen (offensive and defensive) and tight ends exhibit longer inspiratory durations (ID) and higher SMIP compared to wide receivers and cornerbacks.

This positional divergence in inspiratory endurance likely reflects functional adaptations to chronic loading patterns. Linemen and tight ends experience sustained high intra-abdominal and intrathoracic pressure demands during play, particularly during engagement at the line of scrimmage, where the diaphragm and accessory inspiratory muscles act not only in ventilation but also as core stabilizers. The diaphragm, in conjunction with the transversus abdominis and pelvic floor, contributes to the IAP, which enhances spinal stiffness and force transmission during heavy lifting and collision [17]. Repeated exposure to such demands may induce hypertrophy and improved fatigue resistance in the inspiratory musculature, analogous to skeletal muscle adaptations observed in resistance-trained athletes. Indeed, SMIP, as a measure of pressure sustained over time (i.e., area under the pressure-time curve), integrates both strength and endurance components, explaining why linemen, despite similar peak MIP values, outperform wide receivers and defensive backs in sustained respiratory work capacity.

Supporting this interpretation, Hodges and Gandevia demonstrated that the diaphragm contributes up to 60% of tidal volume during quiet breathing but is also recruited during postural tasks requiring trunk stabilization [18]. In football linemen, who routinely generate near-maximal trunk stiffness on every snap, chronic co-activation of respiratory and postural motor units may enhance neural drive and muscle endurance in the diaphragm and intercostals. This is consistent with our observation that defensive linemen and linebackers, positions characterized by explosive starts, violent collisions, and sustained pushing/pulling, showed higher SMIP values, while wide receivers and defensive backs, who experience lower collision frequency and more intermittent respiratory demand, served as the groups with lower values. Of note, the tight end in American football is a hybrid position, aligning adjacent to offensive linemen to provide run-blocking support like offensive linemen, while also detaching into pass routes as an eligible receiver to catch passes, blending the physicality of linemen with the route-running skills of wide receivers, had the highest mean SMIP and ID performances. Running backs and linebackers also had higher SMIP and ID values than wide receiver and defensive backs, as they share similar physiologic demands in football, requiring explosive acceleration, agility, and power for short bursts, combined with high anaerobic capacity and muscular endurance to sustain repeated high-intensity efforts during runs, tackles, and coverage plays.

Furthermore, the longer ID observed in linemen and tight ends may reflect enhanced respiratory muscle stamina, potentially conferring a performance advantage during repeated high-intensity efforts. Football play lasts approximately 5–7 s per snap, with 30–40 s of recovery. Linemen and tight ends, who participate in nearly every offensive and defensive down, accumulate greater total work and ventilatory demand over a game [19]. Superior inspiratory endurance may delay the onset of respiratory muscle metaboreflex activation, a phenomenon where fatigued inspiratory muscles trigger vasoconstriction in locomotor muscles to prioritize ventilatory blood flow [20,21]. By mitigating this metaboreflex, linemen and tight ends with higher SMIP and ID may sustain leg power output longer during late-game drives, a hypothesis warranting future field-based investigation.

Thus, our findings suggest a functional linkage between core stability demands and respiratory muscle endurance in football, particularly among linemen and tight ends. This has implications for position-specific training, while skill players may benefit most from inspiratory muscle training (IMT) to enhance recovery between sprints, linemen and tight ends, despite higher baseline SMIP, may still gain from targeted inspiratory endurance protocols to further delay fatigue under cumulative game stress. Future studies should incorporate electromyography (EMG) of the diaphragm and transversus abdominis during football-specific tasks to directly quantify co-activation patterns and validate this proposed core-respiratory axis.

### 4.2. Inspiratory Muscle Training

Lorca-Santiago et al. conducted a comprehensive systematic review and meta-analysis of IMT in intermittent-sport athletes, synthesizing 21 randomized controlled trials across sports such as rugby, soccer, basketball, and handball [21]. Their pooled analysis revealed that IMT at 40–80% of MIP, performed 3–6 times per week for 4–12 weeks, significantly improved MIP (SMD = 1.12), time-to-exhaustion (SMD = 0.78), and repeated-sprint ability (SMD = 0.65), with moderate-to-large effect sizes. Crucially, these performance gains were mediated by three interconnected physiological mechanisms: (1) reduced perception of dyspnea during high-intensity efforts, (2) delayed onset of respiratory muscle fatigue, and (3) attenuation of the respiratory muscle metaboreflex. This interpretation remains speculative and should be confirmed through experimental studies.

This triad of mechanisms is highly relevant to American football, a sport characterized by intermittent, high-intensity efforts (5–7 s per play) separated by short recovery intervals (25–40 s), during which ventilatory demand remains elevated due to elevated catecholamine levels and post-exercise oxygen consumption [19]. The respiratory metaboreflex, first described by Harms et al. and colleagues, occurs when diaphragmatic fatigue triggers group III/IV afferent feedback, inducing sympathetic vasoconstriction in locomotor muscles to redirect blood flow toward the respiratory pump [22]. This reflex is particularly pronounced in sports requiring repeated maximal efforts, such as football, where inspiratory muscle work can exceed 70% of maximal capacity during play [20]. By increasing MIP and inspiratory muscle endurance, IMT raises the fatigue threshold of the diaphragm, delaying metaboreflex activation, and preserving limb blood flow and power output during late-game scenarios.

Our findings strongly align with and contextualize these mechanisms within collegiate football. Although we did not implement IMT, our observation of higher SMIP and longer ID in linemen, positions with the greatest cumulative ventilatory and metabolic demand. suggests a natural adaptation that mirrors the training-induced changes reported by Lorca-Santiago et al. Specifically, SMIP, as an integrative measure of inspiratory work capacity (pressure × time), captures the same endurance domain that IMT targets. The elevated SMIP in linemen may reflect chronic respiratory muscle loading from repeated Valsalva-like maneuvers during blocking and tackling, akin to a “natural IMT” stimulus. This adaptation could confer a functional buffer against metaboreflex-induced fatigue, allowing linemen to maintain blocking power and push-pull strength over 60–80 snaps per game

IMT has been shown to improve respiratory muscle strength, reduce ventilatory fatigue, and enhance exercise tolerance in various athletic populations [[8][23],[24],[25],[26],[27]]. A systematic review by Santos et al. concluded that IMT using threshold devices significantly improved MIP, metabolic function, and sports performance across disciplines such as sprinting and swimming, and rugby [8]. These improvements are attributed to increased diaphragm hypertrophy, enhanced oxygen delivery, and reduced dyspnea during exertion. The emphasis on acute inspiratory muscle warm-up (IMW), typically 2 sets of 30 breaths at 40% MIP, offers a translatable, low-burden intervention for football. Marostegan et al. demonstrated that IMW performed 5–10 min pre-exercise improved peak power and muscle oxygenation during recovery in sprint athletes, without altering lactate or heart rate [28]. In football, where players must rapidly transition from rest to maximal effort, pre-game IMW could prime the inspiratory muscles, reduce early dyspnea, and enhance recovery between plays, particularly for tight ends, linemen, and linebackers, who face the highest snap counts and collision frequency.

Thus, our cross-sectional data, when viewed through the lens of Lorca-Santiago’s meta-analytic framework, provide ecological validation of IMT’s relevance in football and highlight position-specific opportunities for respiratory intervention [22]. We propose that routine IMT (e.g., 30 breaths at 50% MIP, twice daily, 5 days/week) be integrated into off-season and in-season training, particularly for linemen and hybrid positions (e.g., tight ends, edge rushers), to amplify baseline SMIP and ID beyond natural adaptation. Additionally, pre-game IMW should be explored as a standard component of dynamic warm-up protocols, especially in hot/humid environments where ventilatory demand is exacerbated. Future randomized trials in football cohorts should assess whether IMT/IMW improves snap-to-snap power maintenance, tackle completion rate, or fourth-quarter performance; outcomes directly tied to respiratory endurance and metaboreflex suppression.

### 4.3. Limitations and Future Directions

The cross-sectional design limits causal inference. Small subgroup sizes (e.g., quarterbacks) may have reduced power to detect differences, and the study may be underpowered for small effects (e.g., ρ = 0.243 for height vs. ID; post hoc power ≈ 0.65). Small sample sizes in certain positions (e.g., quarterbacks) also limit statistical power and generalizability. The low adjusted R^2^ values suggest unmeasured factors influence inspiratory performance. Potential interaction effects (e.g., position × offense/defense) were not tested due to sample size constraints and should be explored in future studies. Additional limitations include reliance on BMI rather than direct body composition measures (e.g., DXA), small positional subgroups, and lack of training history data. Lastly, BMI and weight do not capture fat-free mass distribution or diaphragm morphology, which may better predict respiratory muscle capacity.

Future research should incorporate IMT protocols to assess impacts on performance. Integrating additional measures (e.g., electromyography) could elucidate biomechanical interplay. Larger samples and longitudinal designs would address power issues and allow for interaction testing. Future studies should consider including larger, multi-team cohorts. Several of our interpretations are speculative and should be confirmed through experimental studies.

These findings have practical implications for strength and conditioning programs, particularly in tailoring inspiratory muscle training (IMT) for positions with high ventilatory and core stability demands (e.g., linemen, tight ends). Coaches and clinicians may consider integrating IMT and inspiratory warm-up protocols to enhance recovery and performance during repeated high-intensity efforts.

## Figures and Tables

**Figure 1 jfmk-10-00470-f001:**
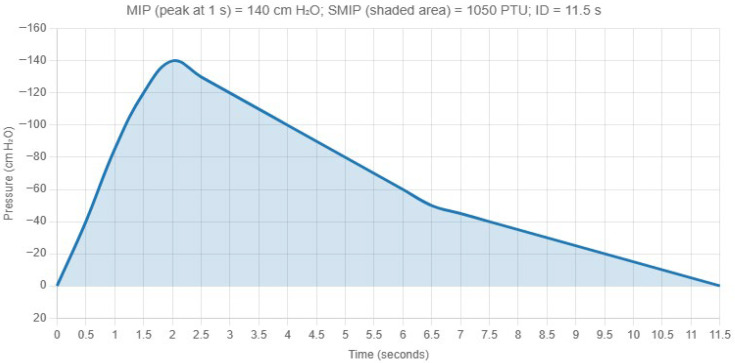
Example of pressure-time curve during the Test of Incremental Respiratory Endurance (TIRE), illustrating MIP (Maximal Inspiratory Pressure), SMIP (Sustained Maximal Inspiratory Pressure), and ID (Inspiratory Duration).

**Table 1 jfmk-10-00470-t001:** Anthropometric and Inspiratory Characteristics of the Participants.

Variable	Min	Max	Mean
Height (in)	68	81	74.3
Weight (kg)	74.84	151.95	108.13
BMI	21.87	40.78	30.21
Year in School	1	6	2.87
MIP (cmH_2_O)	46	187	110.76
SMIP (PTU)	12.30	1571	651.69
ID (s)	3.60	26.60	11.41

MIP—Maximal Inspiratory Pressure (cmH_2_O); SMIP—Sustained Maximal Inspiratory Pressure (PTU); ID—Inspiratory Duration (s).

**Table 2 jfmk-10-00470-t002:** Inspiratory Muscle Performance by Positional Group.

Position (n)	MIP (cmH_2_O)	SMIP (PTU)	ID (s)
QB (5)	127.4	700.9	11.6
RB (6)	116.6	807.5	12.1
WR (11)	107.9	512.5	8.7
TE (11)	104.4	732.2	13.5
OL (13)	110.9	720	13.6
DL (15)	120.5	713.9	11
LB (12)	115.5	703.6	10.7
DB (12)	94.7	452	9.6

MIP—Maximal Inspiratory Pressure (cmH_2_O); SMIP—Sustained Maximal Inspiratory Pressure (PTU); ID—Inspiratory Duration (s); QB—quarterback; RB—running back; WR—wide receiver; TE—tight end; OL—offensive lineman; DL—defensive lineman; LB—linebacker; DB—defensive back.

**Table 3 jfmk-10-00470-t003:** Results of Correlation Analyses with Inspiratory Muscle Performance and Height, Weight, and BMI.

Predictor	Inspiratory Metric	Spearman ρ (*p*-Value)	Interpretation
Height	MIP	0.080 (0.463)	No correlation
Height	SMIP	0.148 (0.175)	No correlation
Height	ID	0.243 (0.024)	Weak positive correlation
Weight	MIP	0.141 (0.197)	No correlation
Weight	SMIP	0.122 (0.262)	No correlation
Weight	ID	0.145 (0.184)	No correlation
BMI	MIP	0.122 (0.263)	No correlation
BMI	SMIP	0.077 (0.480)	No correlation
BMI	ID	0.068 (0.531)	No correlation

MIP—Maximal Inspiratory Pressure (cmH_2_O); SMIP—Sustained Maximal Inspiratory Pressure (PTU); ID—Inspiratory Duration (s); BMI—body mass index.

**Table 4 jfmk-10-00470-t004:** Correlations with Year in School.

Predictor	Inspiratory Metric	Spearman ρ (*p*-Value)	Interpretation
Year	MIP	0.202 (0.062)	No correlation
Year	SMIP	0.170 (0.117)	No correlation
Year	ID	0.090 (0.409)	No correlation

MIP—Maximal Inspiratory Pressure (cmH_2_O); SMIP—Sustained Maximal Inspiratory Pressure (PTU); ID—Inspiratory Duration (s).

**Table 5 jfmk-10-00470-t005:** Results Comparing Inspiratory Muscle Performance between Offensive and Defensive Players.

Inspiratory Metric	Test	Statistic (*p*-Value)	Cohen’s d	Interpretation
MIP	*t*-test	−0.082 (0.935)	0.02	No difference
SMIP	Mann–Whitney	1027.5 (0.338)	0.18	No difference
ID	Mann–Whitney	1077.0 (0.165)	0.23	No difference

MIP—Maximal Inspiratory Pressure (cmH_2_O); SMIP—Sustained Maximal Inspiratory Pressure (PTU); ID—Inspiratory Duration (s).

## Data Availability

The datasets presented in this article are not readily available because the data are part of an ongoing study.

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
