# Peer review of "Inspiratory Muscle Performance and Its Correlates Among Division I American Football Players"

_jfmk, 2025, doi:10.3390/jfmk10040470_

Round 1

Reviewer 1 Report

Comments and Suggestions for Authors

comments in the attachment

Author Response

We would like to thank the reviewer for the comments and recommendations for the manuscript. We believe the quality of the manuscript has markedly improved with the edits. This reply includes the comment section where the recommendation was made, followed by the changes to the document.

Comment 4: Originality and practical usefulness

Lines 378-382: These findings have practical implications for strength and conditioning programs, particularly in tailoring inspiratory muscle training (IMT) for positions with high ventilatory and core stability demands (e.g., linemen, tight ends). Coaches and clinicians may consider integrating IMT and inspiratory warm-up protocols to enhance recovery and performance during repeated high-intensity efforts.

Comment 5: Title and keywords

Inspiratory Muscle Performance and Its Correlates Among Division I American Football Players

inspiratory muscle strength; Division I football physiology; anthropometrics; respiratory endurance; TIRE protocol; positional differences

Comment 8: Methodological assessment

Line 146-149: Given the exploratory nature and multiple comparisons (~30 tests), p-values were unadjusted. Readers should interpret borderline results (e.g., p = 0.024) cautiously, as they may not remain significant after correction. MIP was normally distributed and compared using t-tests, while SMIP and ID were analyzed with non-parametric tests.

Lines 87-90: Positional classification was based on the official team depth chart at the time of testing. Due to small sample sizes in certain positions (e.g., quarterbacks), findings for these groups should be interpreted with caution.

Comment 9: Data analysis and conclusions

Lines 222-225: Regression models yielded low R² values, indicating that unmeasured factors such as training history, respiratory conditioning, and genetic predispositions likely play a major role in inspiratory performance.

Comment 10: Discussion

Lines 212-213: This study found limited associations between inspiratory performance and anthropometric variables, with only a weak correlation between height and inspiratory duration. Positional differences were modest, with linemen and tight ends showing higher SMIP and ID compared to skill positions.

Reviewer 2 Report

Comments and Suggestions for Authors

General comments

This manuscript aims to investigate the relationships between inspiratory muscle performance and anthropometric/positional variables in NCAA Division I collegiate American football players. From a practical point of view, this work has potential implications for position-specific conditioning, inspiratory muscle training strategies, and RTP screening in a high-risk collision sport, such American football. Overall, the study appears interesting and well-structured. However, some issues should be acknowledged and addressed in order to better judge the manuscript. For instance, the statistical approach needs clearer justifications and more details. Additionally, the sample description is limited and some missing methodological details, such as rest intervals, trial selection, are missing. Here below some specific comments.

Specific comments:

Titles: I suggest replacing "among" to "in" and add "American" before "football players". Moreover, I would delete "cross-sectional study", which is specified later in the methods.

Abstract

Rephrase the sentence "may vary by experience and position, suggesting potential benefits from targeted respiratory training in football." as it appears as overstating due to the non-significant results observed for year in school or position-based differences. Please apply such changes also in the Results and Conclusions sections.

Introduction

The authors are invited to better connect respiratory aspects with the position-specific anthropometric and performance literature at the beginning of this section.

Please clearly state that BMI may be an imperfect proxy for body composition in the context of playing position (i.e, linemen often exceed the classic BMI range)

Within the sentence, "This suggests an optimal BMI range for respiratory efficiency,...." I would reinforce by referencing to running-based team-sport o other invasion sport, in which BMI often poorly reflects underlying fat-free mass and fat mass distribution. Here below the potential citation:

10.1186/s12967-023-04795-z

"The research team predicts..." sounds slightly informal. I would also use "We hypothesize" instead "The team..".

Methods

Please clarify how many predictors entered each model

15 predictors are accurate? Clearly depict this number.

Why the authors did not report sex and age of participants?

Please better describe the warm up for replication. For example, specify what exactly means "light stretching and mobility exercise"

Rest intervals between trials are not described for TIRE protocol. What was the stopping rule?

Statistical analyses

Explicitly explain that MIP was normally distributed and therefore compared using t-test rather than non-parametric approach as prioritized before.

Regarding multivariate regression, the authors mentioned exclusion of BMI due to multicollinearity. However, they do not report any diagnostics beyond “VIF >10”, nor how many categorical levels for position were retained. Please add details

Discussion

The authors reported that multivariate regression models identified modest predictors. This particularly among linemen and more experienced players. However, it appears not clearly detailed in the Results.

In terms of metaboreflex and late-game performance, I believe that the link from higher SMIP/id to improved the late-game leg power is understandable, but remain a speculation that should be acknowledged.

The limitations of the study are clearly reported. However, what about the use of BMI rather than more precise body composition measures (i.e, BIA?)

Author Response

We would like to thank the reviewer for their comments. We believe that the comments and recommendations have substantially improved the manuscript. We have made the following edits to the manuscript.  

Title: Inspiratory Muscle Performance and Its Correlates Among Division I American Football Players

Abstract: Inspiratory performance appears largely independent of anthropometric and positional factors; future research should explore targeted respiratory training as a hypothesis rather than a confirmed benefit.

Introduction:

Lines 47-49: The ability to recruit higher-force diaphragm motor units efficiently may contribute to better oxygen delivery and reduced fatigue, especially in linemen during repeated sprints or tackles, as well as core strength.

Lines 80-83: We hypothesized that inspiratory performance would correlate with height and weight, vary by position, and improve with years in college. We also anticipated a negative correlation with BMI, acknowledging BMI as an imperfect proxy for body composition in football players.

Method:

Line 87: Eighty-five male D1CFP...

Age was not included as they are all collegiate aged athletes, between the ages of 18-23.

Lines 110-112: Participants completed a standardized 10-min warm-up consisting of dynamic hamstring and quadriceps stretches, trunk rotations, and shoulder mobility drills.

Lines 124-125: Participants rested 60–90 seconds between trials; testing stopped after three successful trials or participant fatigue.

Statistical analysis:

Lines 148-149: MIP was normally distributed and compared using t-tests, while SMIP and ID were analyzed with non-parametric tests.

Lines 161-166: For multivariate analyses, three separate multiple linear regression models were constructed for MIP, SMIP, and ID as dependent variables. To avoid multicollinearity (variance inflation factors > 10 for BMI due to its derivation from height and weight), BMI was excluded as a predictor. Each model included 11 predictors: height, weight, year in school (continuous), offense/defense status (binary: 0 = offense, 1 = defense), and seven dummy coded position categories (with quarterback as reference).

Lines 169-171: Variance inflation factors (VIF) were computed; BMI was excluded due to VIF >10. Position was dummy-coded into 7 levels.

Discussion:

Lines 222-226: Regression models yielded low R² values, indicating that unmeasured factors such as training history, respiratory conditioning, and genetic predispositions likely play a major role in inspiratory performance.

Lines 368-372: Additional limitations include reliance on BMI rather than direct body composition measures (e.g., DXA), small positional subgroups, and lack of training history data. Lastly, BMI and weight do not capture fat-free mass distribution or diaphragm morphology, which may better predict respiratory muscle capacity. 

Lines 377-378: Several of our interpretations are speculative and should be confirmed through experimental studies.

Reviewer 3 Report

Comments and Suggestions for Authors

This study is relatively new, as inspiratory muscle endurance has been relatively little studied. However, it is well known that aerobic and anthrobic endurance are not limited by respiratory muscle performance, as they are determined by other external and internal factors.

The study was conducted qualitatively, using quantitative physiological methods.

Notes:

1. Abbreviations in tables should be expanded, and units of measurement should be added in parentheses or separated by commas.

2. The figure is missing a caption.

3. In our opinion, the discussion does not sufficiently explain the authors' explanation for the low correlation between anthropometric indicators and inspiratory endurance.

Author Response

We would like to thank the reviewer for their comments and recommendations. We agreed with all three of the recommendations. Below are the edits to the manuscript that were added:

Comment 1:We have improved the labels and expanded the tables.

Comment 2: We have added a caption for the figure.

Comment 3:

Lines 214-227: This study found limited associations between inspiratory performance and anthropometric variables, with only a weak correlation between height and inspiratory duration. Positional differences were modest, with linemen and tight ends showing higher SMIP and ID compared to skill positions.. Bivariate analyses revealed limited associations between inspiratory performance and height, but not weight, BMI, or year in school. Additionally, no statistically significant differences in inspiratory performance were found between position groups. Multivariate regression models identified modest predictors of inspiratory metrics, particularly among linemen and more experienced players. These findings suggest that inspiratory performance may not be inherently tied to anthropometric traits or positional classification alone, but rather influenced by training history, neuromuscular control, and acute physiological interventions. Regression models yielded low R² values, indicating that unmeasured factors such as training history, respiratory conditioning, and genetic predispositions likely play a major role in inspiratory performance.

Lines 363-384: The cross-sectional design limits causal inference. Small subgroup sizes (e.g., quarterbacks) may have reduced power to detect differences, and the study may be underpowered for small effects (e.g., ρ = 0.243 for height vs. ID; post-hoc power ≈ 0.65). Small sample sizes in certain positions (e.g., quarterbacks) also limit statistical power and generalizability. The low adjusted R² values suggest unmeasured factors influence inspiratory performance. Potential interaction effects (e.g., position × offense/defense) were not tested due to sample size constraints and should be explored in future studies. Additional limitations include reliance on BMI rather than direct body composition measures (e.g., DXA), small positional subgroups, and lack of training history data. Lastly, BMI and weight do not capture fat-free mass distribution or diaphragm morphology, which may better predict respiratory muscle capacity. 

Future research should incorporate IMT protocols to assess impacts on performance. Integrating additional measures (e.g., electromyography) could elucidate biomechanical interplay. Larger samples and longitudinal designs would address power issues and allow for interaction testing. Future studies should consider including larger, multi-team cohorts. Several of our interpretations are speculative and should be confirmed through experimental studies.

These findings have practical implications for strength and conditioning programs, particularly in tailoring inspiratory muscle training (IMT) for positions with high ventilatory and core stability demands (e.g., linemen, tight ends). Coaches and clinicians may consider integrating IMT and inspiratory warm-up protocols to enhance recovery and performance during repeated high-intensity efforts.